# Lipid flippases ATP9A and ATP9B form a complex and contribute to the exocytic pathway from the Golgi

Tsukasa Yagi*, Riki Nakabuchi*, Yumeka Muranaka, Gaku Tanaka, Yohei Katoh⬥, Kazuhisa Nakayama⬥,
Hiroyuki Takatsu⬥, Hye-Won Shin⬥

**Type IV P-type ATPases (P4-ATPases) serve as lipid flippases, translocating membrane lipids from the exoplasmic (or luminal) leaflet to the cytoplasmic leaflet of lipid bilayers. In mammals, these P4-ATPases are localized to distinct subcellular compartments. ATP8A1 and ATP9A, members of the P4-ATPase family, are involved in endosome-mediated membrane trafficking, although the roles of P4-ATPases in the exocytic pathway remain to be clarified. ATP9A and ATP9B are located in the TGN, with ATP9A also present in endosomal compartments. This study revealed the overlapping roles of ATP9A and ATP9B in transporting VSVG from the Golgi to the plasma membrane within the exocytic pathway. Furthermore, we demonstrated that the flippase activities of ATP9A and ATP9B were crucial for the transport process. Notably, we discovered the formation of homomeric and/or heteromeric complexes between ATP9A and ATP9B. Therefore, ATP9A and ATP9B play a role in the exocytic pathway from the Golgi to the plasma membrane, forming either homomeric or heteromeric complexes.**

## Introduction

Membrane traffic is facilitated by vesicular and tubular carriers, which are formed at the plasma membrane or organelles through the activities of numerous factors. These include small GTPases, coat proteins, SNARE proteins, fission proteins such as members of the dynamin family, and other accessory proteins. During the formation of these vesicular-tubular carriers and the fusion of organelles, dynamic changes in membrane shape occur. These shape changes are believed to involve alterations in the composition and distribution of lipids, as well as many of the aforementioned proteins (Graham, 2004; Graham & Kozlov, 2010; Shin et al, 2012).

P4-ATPases, a subfamily of P-type ATPases, have been implicated in the flipping of membrane lipids from the exoplasmic (luminal) leaflet to the cytoplasmic leaflet (Palmgren & Nissen, 2011). P4-ATPases are crucial for regulating the trans-bilayer lipid asymmetry of biological membranes: phosphatidylserine (PS) and phosphatidylethanolamine (PE) are primarily located in the cytoplasmic leaflet, whereas phosphatidylcholine (PC), sphingomyelin, and glycolipids are enriched in the exoplasmic leaflet (Op den Kamp, 1979; Devaux, 1991; Zachowski, 1993; Murate et al, 2015). Among P4-ATPases, ATP8B1, ATP8B2, and ATP10A preferentially flip PC and phosphatidylinositol (Takatsu et al, 2014; Naito et al, 2015; Muranaka et al, 2024), ATP11A and ATP11C flip PS and PE (Yabas et al, 2011; Takatsu et al, 2014; Takada et al, 2015), and ATP8A1 and ATP8A2 primarily flip PS (Coleman et al, 2009; Lee et al, 2015). Furthermore, ATP10B flips PC and glucosylceramide, and ATP10D specifically flips glucosylceramide (Roland et al, 2019; Martin et al, 2020). The enhanced phospholipid flipping by ATP10A at the plasma membrane promotes plasma membrane deformation and induces endocytosis (Takada et al, 2018). It is also associated with plasma membrane dynamics, including changes in cell shape (Naito et al, 2015; Miyano et al, 2016). In addition, phospholipid flipping itself, as opposed to the flipped phospholipids, is important for vesicle formation in yeast (Takeda et al, 2014). Therefore, local changes in the composition of specific phospholipids by P4-ATPases are actively involved in the process of membrane deformation.

All five yeast P4-ATPases participate in protein transport at various stages of the exocytic and endocytic pathways (Muthusamy et al, 2009). Some yeast P4-ATPases interact with proteins implicated in vesicle formation, such as ADP-ribosylation factor-guanine nucleotide exchange factors (ARF-GEFs). These factors activate small GTPases, which are involved in the recruitment of coat proteins and cargo receptors to vesicles (Wicky et al, 2004; Natarajan et al, 2009; Tsai et al, 2013). P4-ATPases also play crucial roles in membrane trafficking in other organisms, including *Caenorhabditis elegans* and *Arabidopsis thaliana* (Poulsen et al, 2008; Anne-Francoise et al, 2009; Lopez-Marques et al, 2021). In mammals, ATP8A1 is necessary for the recruitment of EHD1 to endosomes and

Graduate School of Pharmaceutical Sciences, Kyoto University, Kyoto, Japan

Correspondence: shin@pharm.kyoto-u.ac.jp
Yohei Katoh's present address is Hiroshima University Genome Editing Innovation Center, Hiroshima, Japan
*Tsukasa Yagi and Riki Nakabuchi contributed equally to this work

for endosome-mediated trafficking in COS-1 cells (Lee et al, 2015). ATP9A is involved in the recycling pathway from endosomes to the plasma membrane of the transferrin receptor and glucose transporter 1 (Tanaka et al, 2016). ATP9A also plays a role in the recycling of Wntless and Wnt secretion (McGough et al, 2018). Therefore, the phospholipid flipping activities of P4-ATPases are vital for membrane trafficking in eukaryotic cells.

The majority of mammalian P4-ATPases necessitate an association with CDC50 for their exit from the ER and subsequent cellular localization (Paulusma et al, 2008; Bryde et al, 2010; van der Velden et al, 2010; Coleman & Molday, 2011; Takatsu et al, 2011; Lopez-Marques et al, 2014). However, ATP9A and ATP9B, considered mammalian orthologs of yeast Neo1p, do not require interaction with CDC50 for ER-exit (Saito et al, 2004; Takatsu et al, 2011; Tone et al, 2020).

Early studies highlighted the involvement of yeast and plant P4-ATPases in the post-Golgi exocytic pathway (Gall et al, 2002; Hua et al, 2002; Chen et al, 2006; Poulsen et al, 2008), but the involvement of mammalian P4-ATPases in the exocytic pathway remains unknown. In this study, we examined the function of ATP9A and ATP9B, which localize to the TGN, in the post-Golgi exocytic pathway. We discovered that ATP9A and ATP9B redundantly contribute to the exocytic pathway from the Golgi apparatus to the plasma membrane. Notably, their flippase activities appeared to be essential for protein transport. Furthermore, we demonstrated that ATP9A and ATP9B function as homomeric and/or heteromeric complexes in the transport pathway.

# Results

### VSVG transport from the Golgi to the plasma membrane is delayed in ATP9A or ATP9B KO cells

The fact that ATP9A is localized to the TGN and early/recycling endosomes and ATP9B is localized to the TGN (Fig S1A and B) (Takatsu et al, 2011; Tone et al, 2020) prompted us to investigate their roles in post-Golgi membrane trafficking. To this end, we established *ATP9A*-KO and *ATP9B*-KO HeLa cells using the CRISPR/Cas9 system (Tanaka et al, 2016) (Fig S2) and transfected these KO cells with the N-terminally EGFP-tagged temperature-sensitive glycoprotein of the vesicular stomatitis virus (VSVG-tsO45) (Nishimoto-Morita et al, 2009). The cells were incubated at 40°C overnight to accumulate the EGFP-VSVG-tsO45 (EGFP-VSVG) in the ER, then shifted to 32°C to allow its exit from the ER for transport to the plasma membrane (Fig 1A and B). To inhibit protein synthesis during the 32°C incubation, the cells were treated with cycloheximide.

The EGFP-VSVG protein that reached the plasma membrane was detected by incubating fixed cells with an anti-GFP antibody without permeabilization, followed by an AlexaFluor 555-conjugated secondary antibody (Fig S3A). Thus, the red fluorescence indicates EGFP-VSVG proteins that have reached the cell surface (Fig 1Aa and a'). When cells were incubated at 40°C following transfection with the EGFP-VSVG construct, the EGFP signals were rarely detected, likely due to folding defects (Fig S3B)

(Nishimoto-Morita et al, 2009). After 32°C incubation, the EGFP-VSVG emerged in the Golgi area, followed by the appearance of red fluorescent signals, indicating its transport to the plasma membrane through the Golgi (Fig S3C and D, Control panels).

The red fluorescent signals were considerably reduced in *ATP9A*-KO (clone 23) and *ATP9B*-KO (clone 1) cells compared with parental control cells, suggesting a notable delay in the transport of VSVG to the plasma membrane in these KO cells (Fig 1Ab, b', e, and e'). To circumvent clonal effects, we verified the delay in VSVG transport to the plasma membrane in additional clones of *ATP9A*-KO (clone 2) and *ATP9B*-KO (clone 4) cells (Fig S3B–D). We quantified the fluorescence intensity of EGFP–VSVG on the cell surface (red fluorescent signals), which was detected using an anti-GFP antibody. The intensity was then normalized with the total EGFP intensity to account for variations in the expression level of EGFP–VSVG. The surface VSVG signals were significantly diminished in cells depleted of ATP9A or ATP9B (Fig 1B). To validate the specific roles of ATP9A and ATP9B, we stably expressed C-terminally HA-tagged ATP9A(WT) or ATP9B(WT) in individual KO cells. The delay in VSVG transport to the plasma membrane was rescued by the expression of ATP9A(WT) in *ATP9A*-KO cells and that of ATP9B(WT) in *ATP9B*-KO cells (Fig 1Ac, c', f, and f' and B). The results suggest that ATP9A and ATP9B play a role in the efficient transport of VSVG from the Golgi to the plasma membrane.

Subsequently, we investigated whether the flippase activities of ATP9A and ATP9B are required for VSVG transport to the plasma membrane. To address this, we stably expressed HA-tagged ATP9A(E195Q) or ATP9B(E272Q) mutants, which are ATPase-deficient mutants (Anthonisen et al, 2006; Palmgren & Nissen, 2011; Takatsu et al, 2014). The transport of VSVG to the plasma membrane was partially, but significantly, restored by the expression of the ATPase-deficient mutant, albeit to a slightly lesser extent compared with cells expressing the WT counterpart (Fig 1Ad, d', g, and g' and B). Therefore, a straightforward interpretation could be that the flippase activities of ATP9A and ATP9B are not essential for VSVG transport from the Golgi to the plasma membrane. We confirmed the expression of ATP9A-HA, ATP9A(E195Q)-HA, ATP9B-HA, and ATP9B(E272Q)-HA by immunoblot and immunofluorescence analyses (Fig 1C and Db, c, e, and f).

### The VSVG transport requires the flippase activity of ATP9A and ATP9B

Apart from the possibility that the flippase activities of ATP9A and ATP9B are not essential for VSVG transport, we investigated another potential scenario; ATP9A and ATP9B might work cooperatively by forming a complex. We hypothesized that endogenous ATP9B in *ATP9A*-KO cells could interact with the exogenously expressed ATP9A(E195Q) mutant, resulting in partial mitigation of the delay in VSVG transport to the plasma membrane. To test this hypothesis, we generated *ATP9A* and *ATP9B* double KO (DKO) cells by depleting *ATP9B* in *ATP9A*-KO cells using the CRISPR/Cas9 system (Fig S4). Subsequently, we established DKO cells stably expressing ATP9A(WT), ATP9B(WT), ATP9A(E195Q), or ATP9B(E272Q). As anticipated, the transport of VSVG to the plasma membrane was delayed in the DKO cells (Fig 2Ab and b'). We quantified the cell surface

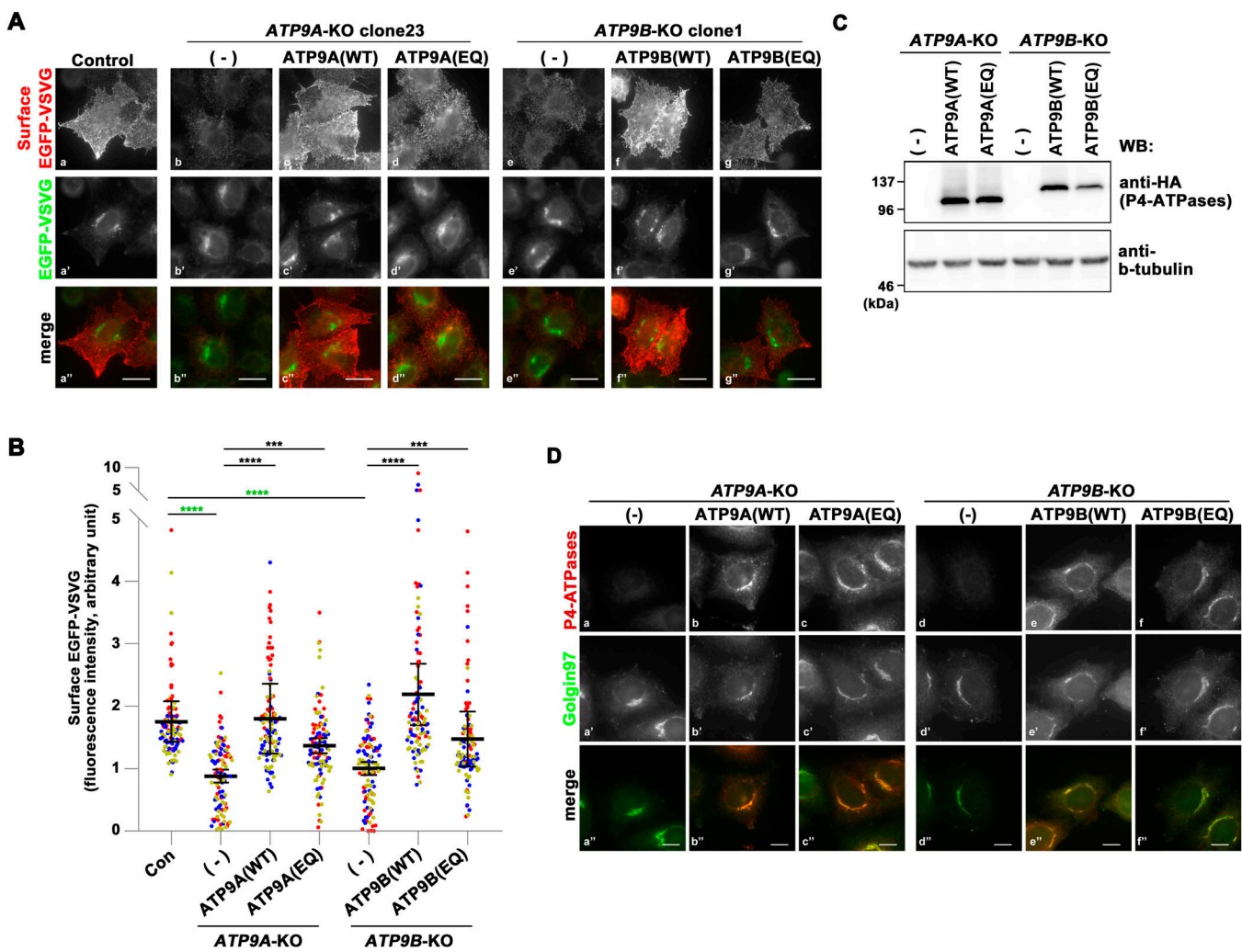

**Figure 1. Transport of VSVG-*ts*O45 in *ATP9A*- and *ATP9B*-KO cells.**
Parental HeLa (control), *ATP9A*-KO, and *ATP9B*-KO cells, along with individual KO cells stably expressing C-terminally HA-tagged ATP9A(WT), ATP9B(WT), ATP9A(EQ) and ATP9B(EQ) mutants, were transfected with an expression vector for N-terminally EGFP-tagged VSVG-*ts*O45. The cells were then incubated at 40°C overnight to accumulate EGFP–VSVG at the ER, followed by a shift to 32°C for 60 min to facilitate the transport of EGFP–VSVG to the plasma membrane. Cycloheximide was added during the incubation at 32°C. **(A)** To detect cell surface expression of EGFP–VSVG, cells were stained with an anti-GFP antibody under non-permeabilized conditions, followed by incubation with AlexaFluor 555-conjugated anti-rabbit secondary antibody (red). The images are representative of three independent experiments. Bars, 20 μm. **(B)** Surface fluorescence intensities (red) were quantified and normalized with EGFP fluorescence intensities (green) using ZEN software. The scatter plot displays the average ± SD from three independent experiments. Variance was assessed using a one-way ANOVA, and comparisons were made with parental HeLa cells (control) (indicated by green asterisks) and with individual KO cells (indicated by black asterisks) using Tukey's post hoc analysis. A total of 99 cells were quantified per sample, with each dot representing a single cell and the colors corresponding to three independent experiments. ****$P < 0.0001$, ***$P < 0.001$. **(C)** The expression of individual ATP9 proteins was confirmed by immunoblotting with anti-HA and anti-$\beta$-tubulin (as an internal control) antibodies. **(D)** Immunofluorescence analysis was performed with anti-HA and anti-Golgin97 antibodies, followed by incubation with Cy3-conjugated anti-rat and AlexaFluor 488-conjugated anti-mouse secondary antibodies. Bars, 10 μm.

fluorescence intensity of EGFP-VSVG as previously described in Fig 1 (Fig 2B). The transport of VSVG to the plasma membrane was restored by the expression of ATP9A(WT) or ATP9B(WT) in the DKO cells (Fig 2Ac', e, and e' and B), suggesting the redundant function of ATP9A and ATP9B. Importantly, the VSVG transport was not restored by the expression of the ATP9A(E195Q) or ATP9B(E272Q) mutant in the DKO cells (Fig 2Ad, d', f, and f' and B), even though the E-to-Q (EQ) mutants could restore the transport in the single KO cells (Fig 1A and B).

We also verified the ER-to-Golgi transport of VSVG using C-terminally EGFP-tagged VSVG (Fig S5), as N-terminally EGFP-tagged

VSVG showed minimal EGFP signals in the ER upon incubation at 40°C due to folding issues (Fig S3B). Given that protein export from the TGN is inhibited at 19.5°C (Griffiths & Simons, 1986), we conducted a time course experiment at this temperature after 40°C incubation. The appearance of VSVG-EGFP in the Golgi apparatus was similar in control and DKO cells, indicating that ER-to-Golgi transport was unaffected in DKO cells (Fig S5).

These findings suggest that (1) the flippase activity of ATP9A or ATP9B is essential for VSVG transport from the Golgi to the plasma membrane, and (2) ATP9A and ATP9B might function in VSVG transport as complexes. The expression of these ATP9 proteins was

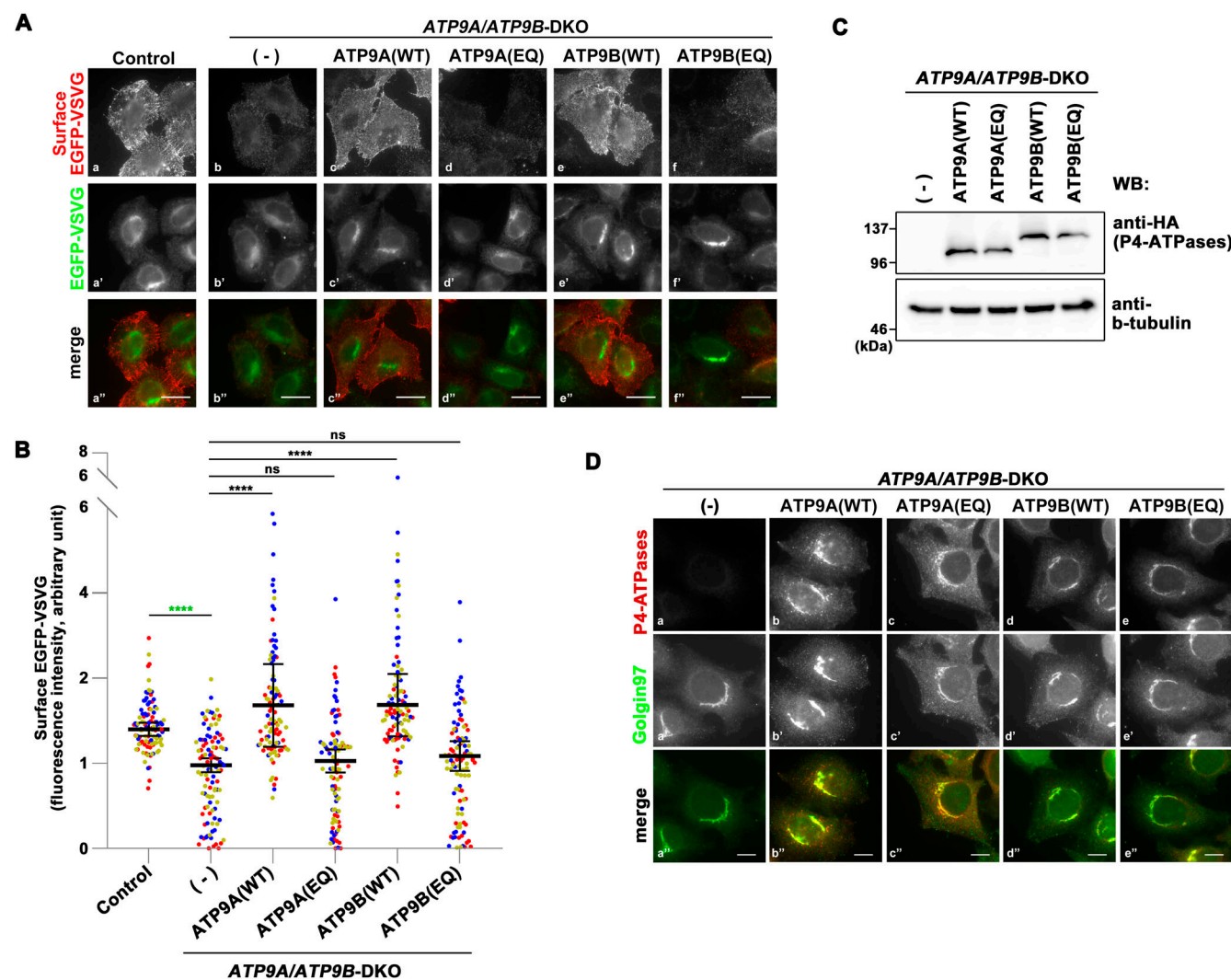

**Figure 2. Transport of VSVG-tsO45 in *ATP9A/ATP9B*-DKO cells.**

Parental HeLa (Control), *ATP9A/ATP9B*-DKO cells, and DKO cells stably expressing C-terminally HA-tagged ATP9A(WT), ATP9A(EQ), ATP9B(WT), and ATP9B(EQ) mutants were transfected with an expression vector for N-terminally EGFP-tagged VSVG-tsO45. The cells were then incubated at 40°C overnight and shifted to 32°C for 60 min to allow the exit from the ER and transport to the plasma membrane of EGFP–VSVG. Cycloheximide was added during the incubation at 32°C. **(A)** To detect cell surface expression of EGFP–VSVG, cells were stained with an anti-GFP antibody under non-permeabilized conditions, followed by incubation with AlexaFluor 555-conjugated anti-rabbit secondary antibody (red). The images are representative of three independent experiments. **(B)** Surface fluorescence intensities (red) were quantified and normalized with EGFP fluorescence intensities (green) using ZEN software. The scatter plot displays the average ± SD from three independent experiments. Variance was assessed using a one-way ANOVA, and comparisons were made with parental HeLa cells (control) (indicated by green asterisks) and with DKO cells (indicated by black asterisks) using Tukey's post hoc analysis. A total of 99 cells were quantified per sample, with each dot representing a single cell, and colors corresponding to three independent experiments. ****$P < 0.0001$, ns, not significant. **(C)** The expression of individual ATP9 proteins in the DKO cells was confirmed by immunoblotting with anti-HA and anti-$\beta$-tubulin (as an internal control) antibodies. **(D)** Immunofluorescence analysis was performed with anti-HA and anti-Golgin97 antibodies, followed by incubation with Cy3-conjugated anti-rat and AlexaFluor 488-conjugated anti-mouse secondary antibodies. Bars, 10 $\mu$m.

validated by immunofluorescence and immunoblot analyses (Fig 2C and D).

## ATP9A and ATP9B form either homomeric or heteromeric complexes

To investigate the complex formation between ATP9A and ATP9B, we established cells that stably coexpress HA-tagged and FLAG-tagged ATP9A and ATP9B, including their EQ mutants, in various combinations (Fig 3A). We also established cells stably expressing

HA-tagged ATP9A or ATP9B, along with FLAG-tagged ATP11C, a P4-ATPase family member, as a negative control. We then performed coimmunoprecipitation analysis on the cell lysates. Exogenously expressed HA- or FLAG-tagged ATP9A and ATP9B were detected as ~110 and ~120 kD bands, respectively, on SDS–PAGE (Fig 3A, total lysate). As depicted in Fig 3A, FLAG-tagged ATP9A, ATP9B, and their EQ mutants were coimmunoprecipitated with ATP9A-HA in cells coexpressing ATP9A-HA and various FLAG-tagged P4-ATPases (lanes 1–4). A similar pattern was observed with ATP9B-HA (lanes 6–9). However, FLAG-tagged ATP11C was not coimmunoprecipitated

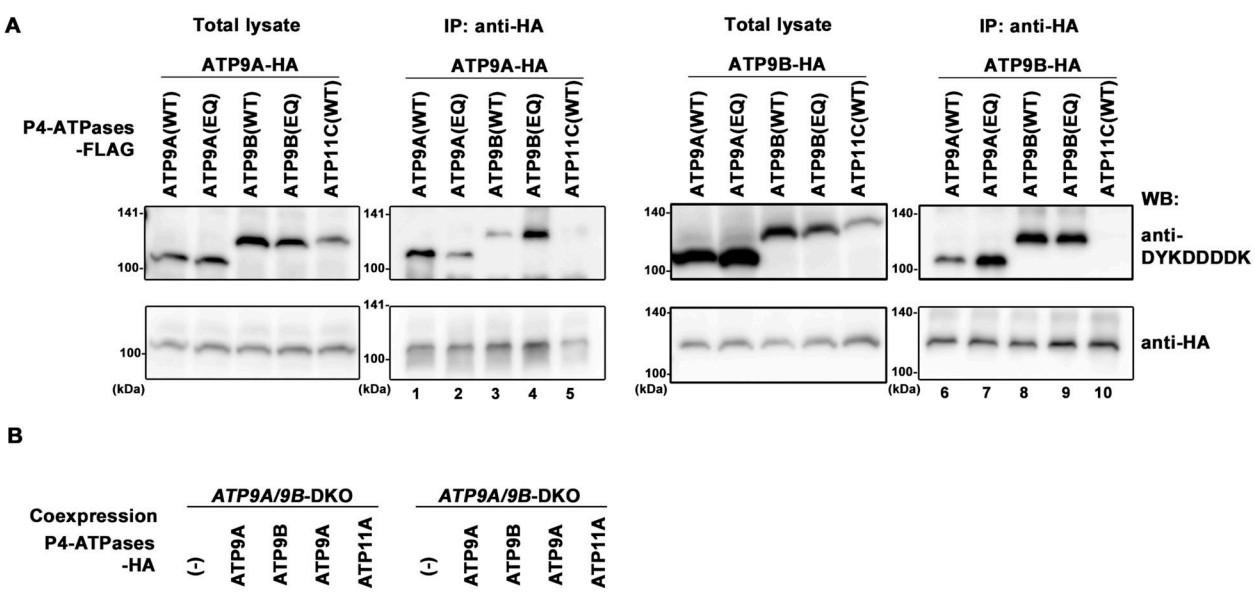

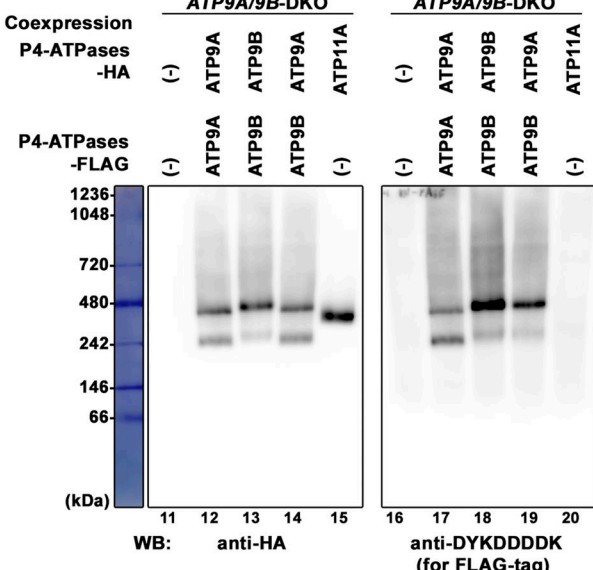

**Figure 3. Formation of a homomeric and/or heteromeric complex between ATP9A and ATP9B.**
**(A)** HeLa cells, stably coexpressing HA-tagged ATP9A with FLAG-tagged ATP9A(WT), ATP9B(WT), ATP9A(EQ), ATP9B(EQ), or ATP11C, as well as HA-tagged ATP9B with FLAG-tagged ATP9A(WT), ATP9B(WT), ATP9A(EQ), ATP9B(EQ), or ATP11C, were lysed. Immunoprecipitation was performed using an anti-HA antibody. Subsequently, total lysates (10% of input) and bound materials were subjected to SDS–PAGE and immunoblotting using anti-HA and anti-DYKDDDDK (for FLAG tag) antibodies. **(B)** *ATP9A*/*ATP9B*-DKO cells, stably coexpressing HA-tagged ATP9s and FLAG-tagged ATP9s in combinations as indicated, or expressing HA-tagged ATP11A alone, were lysed and subjected to BN-PAGE. Immunoblotting was performed using anti-HA and anti-DYKDDDDK antibodies.

with either ATP9A-HA or ATP9B-HA (lanes 5 and 10). These findings suggest that ATP9A and ATP9B form specific homomeric and heteromeric complexes. Moreover, ATPase-deficient mutants (EQ) of ATP9A and ATP9B can also form complexes with ATP9A (lanes 2 and 7) and ATP9B (lanes 4 and 9) (Fig 3A). As previously hypothesized, endogenous ATP9B in *ATP9A*-single KO cells can form heteromeric complexes with the exogenously introduced ATP9A(EQ) mutant, whereas endogenous ATP9A in *ATP9B*-single KO cells can form complexes with the exogenously introduced ATP9B(EQ). Therefore, the exogenous expression of EQ mutants may rescue the defects in *ATP9A*- or *ATP9B*-single KO cells (Fig 1A and B).

Next, we explored the interaction between ATP9A and ATP9B under nondenaturing conditions using blue native PAGE (BN-PAGE) (Fig 3B). To avoid interference from endogenous ATP9A

and ATP9B, we used *ATP9A*/*9B* DKO cells stably coexpressing HA- and FLAG-tagged ATP9A, HA- and FLAG-tagged ATP9B, or HA-tagged ATP9A and FLAG-tagged ATP9B. We also established *ATP9A*/*9B* DKO cells stably expressing ATP11A-HA as a control. After BN-PAGE of the cell lysates, we conducted immunoblot analysis with antibodies for the HA- and FLAG-tags (Fig 3B). Both ATP9A-HA and ATP9B-HA showed migration at ~470~480 and 240 kD with ATP9B-HA slightly higher than ATP9A-HA (lanes 12 and 13). ATP9A-FLAG and ATP9B-FLAG showed similar migration patterns to their HA-tagged counterparts (lanes 17 and 18), suggesting potential homomeric complex formation. ATP11A-HA exhibited migration at ~400 kD, indicating its heteromeric complex formation with CDC50A (lane 15) (Takatsu et al, 2011), similar to ATP11C (Fig S6) (Miyata et al, 2022). ATP9A and ATP9B do

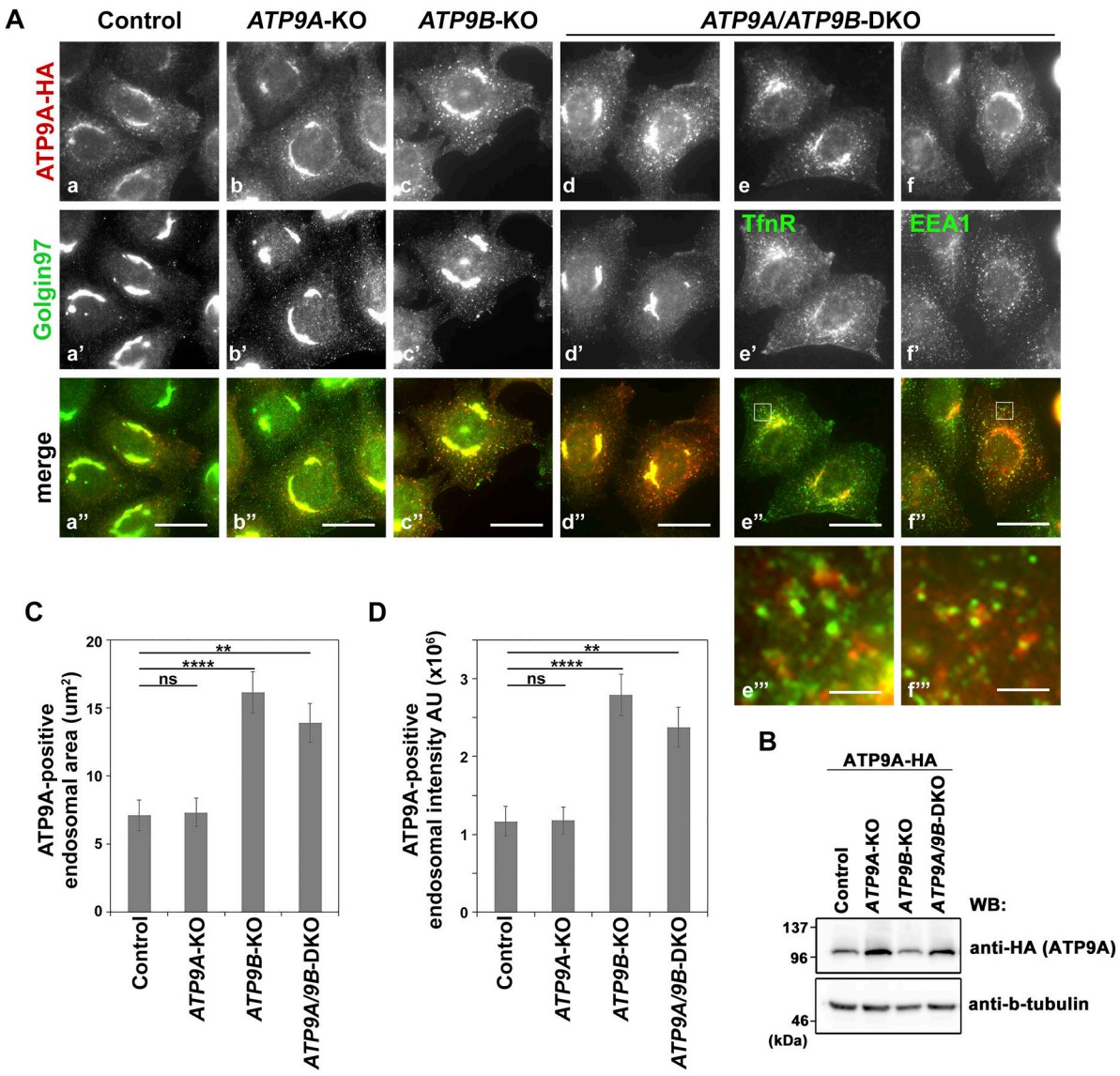

**Figure 4. Changes in ATP9A endosomal localization in cells lacking ATP9B.**
**(A)** Parental HeLa (control), *ATP9A*-KO, *ATP9B*-KO, and *ATP9A/ATP9B*-DKO cells stably expressing HA-tagged ATP9A were fixed, permeabilized, and then incubated with anti-HA and anti-Golgin97 antibodies (a marker for the Golgi apparatus), followed by Cy3-conjugated anti-rat and AlexFlour 488-conjugated anti-mouse secondary antibodies. DKO cells expressing HA-tagged ATP9A were incubated with anti-HA and anti-TfnR or anti-EEA1, followed by Cy3-conjugated anti-rat and AlexFlour 488-conjugated anti-mouse secondary antibodies. Bars, 20 $\mu$m. **(B)** The expression of ATP9A-HA in individual cells were confirmed by immunoblotting with anti-HA and anti-$\beta$-tubulin (as an internal control) antibodies. **(C, D)** The area of ATP9A-positive peripheral vesicular compartments (C) and their fluorescence intensity (D) were quantified using ZEN software. To exclude the ATP9A-positive perinuclear Golgi area, the ROI corresponding to the Golgin97 area was subtracted. The graph represents data from two independent experiments and displays the average ± SE of over 30 cells in each sample. Variance was assessed using a one-way ANOVA, and comparisons were made with parental HeLa cells (control) using Tukey's post hoc analysis. ****$P < 0.0001$, **$P < 0.01$, ns, not significant.

not interact with CDC50A ([Takatsu et al, 2011](); [Tone et al, 2020]()). Moreover, in cells coexpressing ATP9A-HA and ATP9B-FLAG, both ATP9 proteins appeared to be of similar size (lanes 14 and 19), although the upper band of ATP9B-FLAG showed greater intensity than its lower band (lane 19). These findings strongly support our hypothesis that ATP9A and ATP9B form either homomeric or heteromeric complexes. We also confirmed that the migration of ATP11A-HA was not altered in cells coexpressing with ATP9A or ATP9B compared with the control ([Fig S6](), lanes 1–3).

## ATP9B contributes to the localization of ATP9A to the Golgi apparatus

During the above experiments, we unexpectedly noted changes in the cellular localization of ATP9A, which was stably expressed in *ATP9B*-KO and DKO cells, compared with control cells ([Fig 4A]()). Specifically, ATP9A-positive puncta significantly increased in *ATP9B*-KO and DKO cells ([Fig 4Ac–f]()), whereas no significant change was observed in *ATP9A*-KO cells ([Fig 4Ab]()) compared with parental HeLa cells ([Fig 4Aa]()). The expression level of ATP9A was comparable

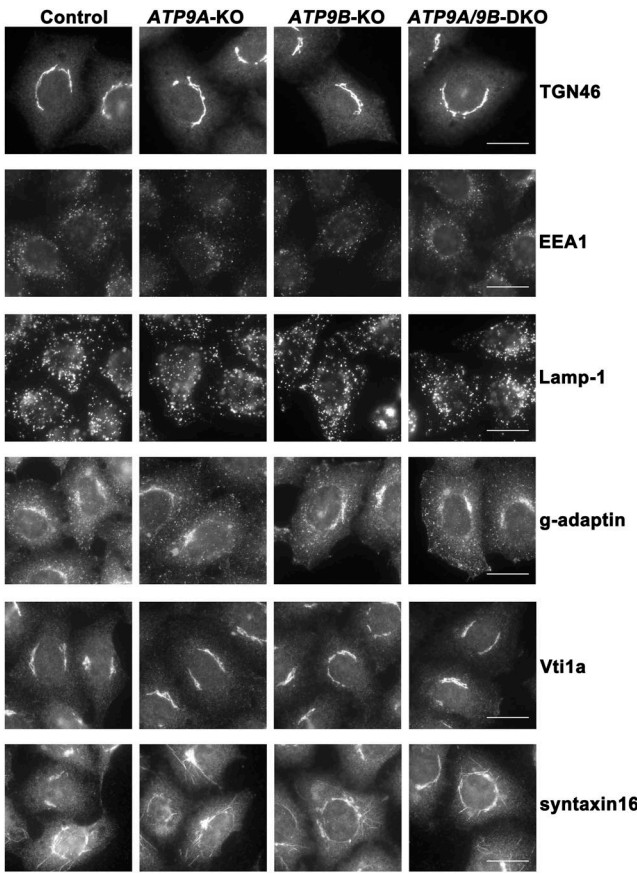

**Figure 5. Unaltered distribution of membrane traffic–related proteins in cells depleted of ATP9A and/or ATP9B.**

Parental HeLa (Control), *ATP9A*-KO, *ATP9B*-KO, and *ATP9A/ATP9B*-DKO cells were fixed, permeabilized, and immunostained with antibodies against TGN46, EEA1, Lamp-1, γ-adaptin, Vti1a, or syntaxin16. Bars, 20 $\mu m$.

across these cells (Fig 4B). We quantified the peripheral ATP9A-positive puncta area and fluorescence intensity using ZEN software (Fig 4C and D). For this, we subtracted the fluorescence signal of ATP9A in the Golgi by excluding the ROI corresponding to the Golgin97-positive area (Fig 4Aa′–d′). The area and fluorescence intensity of ATP9A-positive puncta showed a significant increase in cells lacking ATP9B (*ATP9B*-KO and DKO) (Fig 4C and D). This suggests that ATP9B contributes to either the retention or retrieval of ATP9A at the TGN, further supporting the formation of a heteromeric complex between ATP9A and ATP9B, as shown in Fig 3. Peripheral ATP9A was minimally colocalized with EEA1 and transferrin receptor (TfnR), markers of early and recycling endosomes (Fig 4Ae″, e‴, f″, and f‴). These ATP9A-positive puncta at the periphery may include both endocytic and exocytic vesicular structures.

Considering the observed changes in localization of ATP9A in cells lacking ATP9B, we examined other organelle markers, including Golgi markers. TGN46 serves as a TGN marker, and it travels between the Golgi and the plasma membrane through endosomes (Mallet & Maxfield, 1999). Vti1a and syntaxin 16, SNARE proteins, also cycles between the TGN and the plasma membrane via endosomes (Mallard et al, 2002). γ-adaptin, a component of clathrin adapter protein complex-1 (AP-1), localizes to the TGN and endosomes and

facilitates protein trafficking between these organelles. EEA1 and Lamp-1 serve as markers for early and late endosomes, respectively. None of these markers showed significant changes in *ATP9B*-KO, DKO, and *ATP9A*-KO cells compared with control cells (Fig 5). Thus, it seems that ATP9A and ATP9B are not essential for the recycling of at least TGN46 and some SNARE proteins between the TGN and the plasma membrane, and for the recruitment of TGN-localizing proteins, such as AP-1 (Figs 5 and S7).

## Flippase activities of ATP9A and ATP9B contribute to generation of membrane curvature at the Golgi apparatus

In our previous work, we demonstrated that enhancing flippase activity at the plasma membrane could induce membrane deformation (Takada et al, 2018). This was achieved using a chemically inducible dimerization technique, which enables the acute recruitment of the curvature-sensing BAR domain (specifically, the BAR domain of amphiphysin 1 with the N-terminal amphipathic helix deleted) to the plasma membrane (Komatsu et al, 2010; Suarez et al, 2014; Takada et al, 2018). We transfected HeLa cells with two constructs: an EGFP-tagged FRB (FK506-binding protein [FKBP]-rapamycin-binding domain) fused to GalT (galactosyltransferase) for targeting to the trans-Golgi, and an mCherry-tagged FKBP fused to the BAR domain (Fig S8A) (Suarez et al, 2014; Takada et al, 2018). Golgi morphology was visualized using fluorescence microscopy, with fluorescent signals from the Golgi-targeting FRB-GalT and FKBP-BAR domain following rapamycin treatment, which induces heterodimerization of FRB and FKBP. In the absence of rapamycin, the FKBP-BAR domain was primarily distributed throughout the cytoplasm, and the Golgi-targeting FRB-GalT localized to the Golgi apparatus in control cells (Fig 6Aa–c′). Upon rapamycin treatment, the FKBP-BAR domain was rapidly recruited to the Golgi (Fig 6Ad–i′, 2.5 and 5 min). Notably, after 10 min, FRB-GalT-positive tubular structures emerged from the Golgi apparatus (Fig 6Aj–l′, white arrows), with the FKBP-BAR domain also present in the tubular structures. In addition, small punctate FRB-GalT-positive structures appeared in the cytoplasm (Fig 6Aj″-l″, green arrows). After 15 min, the FRB-GalT-positive tubules became more pronounced (Fig 6Am–o′, white arrows), accompanied by an increase in punctate structures (Fig 6Am″–o″, green arrows). After 30 min, the FRB-GalT signals at the Golgi were reduced, and tubular structures were hardly observed (Fig 6Ap–r′). Instead, punctate structures became more pronounced (Fig 6Ap–r″, green arrows). The FKBP-BAR signal at the Golgi area was dramatically diminished (Fig 6Ap and p′). These observations suggest that the acute recruitment of the BAR domain to the Golgi apparatus promotes tubule formation. The number of FRB-GalT tubulated cells (Golgi tubulated cells) was quantified at each time point (Fig 6B). To distinguish tubular structures from the intricate Golgi stacks, only cells with thin tubules longer than 1 $\mu m$ were included in the analysis (Fig 6A, arrows). As tubules are continuously formed and vesiculated following FKBP-BAR domain recruitment to the Golgi apparatus, the number of tubulated cells decreased over time.

We then transfected DKO cells with the two constructs, FRB-GalT-EGFP and mCherry-FKBP-BAR, and examined FRB-GalT-positive tubule formation (Fig S8B). Upon rapamycin treatment, FKBP-BAR domain was rapidly recruited to the Golgi (Fig S8Bd–f′). After 10 min,

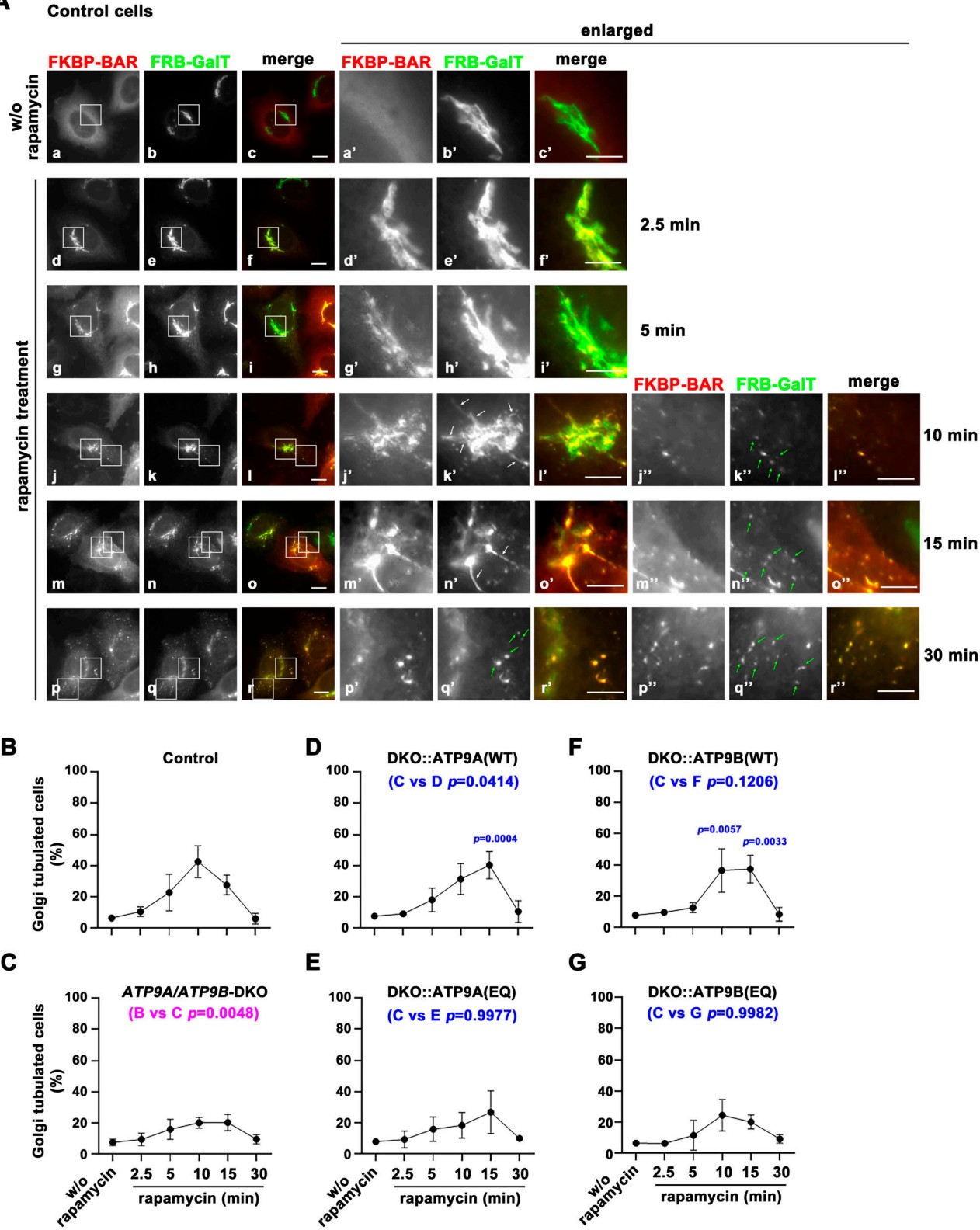

**Figure 6. Tubule formation from the Golgi apparatus via acute recruitment of BAR domain.**
Parental HeLa (control) cells, *ATP9A/ATP9B*-DKO cells, and the DKO cells stably expressing ATP9A(WT), ATP9B(WT), ATP9A(EQ), or ATP9B(EQ) were transiently cotransfected with expression vectors for FRB-GalT-EGFP (FRB-GalT) and mCherry-FKBP-BAR (FKBP-BAR). The cells were treated with either vehicle (w/o rapamycin) or 400 nM rapamycin for the indicated times and then fixed. **(A)** Representative fluorescence microscopy images of control cells are shown. White arrows indicate tubular

FRB-GalT-positive tubular structures were reduced compared with control cells. After 30 min, fragmented Golgi structures appeared in DKO cells, that were hardly observed in control cells (Fig S8Bp–r''). As shown in Fig 6C, the number of tubulated cells was significantly lower in the DKO cells compared with control cells (panel B versus C). The FKBP-BAR domain, which lacks the N-terminal amphipathic helix is deleted, is unable to induce membrane curvature. However, it can still generate tubules when it binds to preexisting curved membranes (Suarez et al, 2014; Takada et al, 2018). Therefore, ATP9A and ATP9B likely play a role in promoting membrane curvature formation at the Golgi apparatus.

To determine whether the flippase activities of ATP9A/9B are required, we transfected DKO cells stably expressing ATP9A(WT), ATP9A(EQ), ATP9B(WT), and ATP9B(EQ) with FRB-GalT and FKBP-BAR constructs (Fig 6D–G). A statistically significant rescue was observed with ATP9A(WT) expression (Fig 6D, panel C versus D), whereas no significant rescue was seen with ATP9B(WT) expression (Fig 6F, panel C versus F). However, ATP9B(WT) expression showed statistically significant rescue effects at some individual time points, at 10 and 15 min (Fig 6F). In contrast, ATP9A(EQ) and ATP9B(EQ) expression showed no significant differences compared with the DKO cells, even at specific time points (Fig 6E and G). These results suggest that the flippase activities of ATP9A and ATP9B at the Golgi apparatus may facilitate membrane deformation, thereby contributing to tubule formation induced by BAR domain recruitment.

## Discussion

In this study, we demonstrated that ATP9A and ATP9B, localized to the TGN, are essential for the efficient transport of VSVG from the Golgi to the plasma membrane. Despite the integrity of cellular compartments, including the Golgi apparatus, remaining intact in both single KO and DKO cells, VSVG transport was delayed in the KO cells. Specifically, (1) the delay in VSVG transport was observed in both *ATP9A*-KO and *ATP9B*-KO cells. (2) Expression of either ATP9A or ATP9B in the DKO cells successfully restored the delayed VSVG transport. These findings suggest a redundant function of ATP9A and ATP9B in facilitating VSVG transport from the Golgi to the plasma membrane.

Interestingly, we observed that the expression of ATP9A(EQ) and ATP9B(EQ) mutants partially rescued the delayed VSVG transport in single KO cells but not in DKO cells. This observation leads to two key insights: first, the flippase activity of either ATP9A or ATP9B is crucial for efficient VSVG transport. Second, ATP9A and ATP9B likely operate as a heteromeric complex. Indeed, ATP9A and ATP9B can form homomeric and heteromeric complexes. We previously demonstrated that enhanced flipping activity at the plasma membrane, induced by ATP10A expression, increases membrane

curvature and promotes integrin endocytosis (Naito et al, 2015; Takada et al, 2018). In yeast, lipid flipping activity is necessary for vesicle transport (Takeda et al, 2014). Thus, the flipping activities of ATP9A and ATP9B likely contribute to membrane deformation at the TGN, facilitating efficient vesicle formation at the TGN. Indeed, BAR domain-induced membrane tubulation from the Golgi apparatus was significantly reduced in DKO cells. However, the expression of ATP9A or ATP9B in these cells restored the tubule formation, whereas their EQ mutants failed to restore it (Fig 6).

While ATP9A and ATP9B can form a homomeric complex in *ATP9B*-KO and *ATP9A*-KO cells, respectively, VSVG transport to the plasma membrane was delayed in these KO cells. This suggests that the heteromeric complex may be more efficient in facilitating VSVG transport. Another possibility is that the endogenous levels of ATP9A or ATP9B may change in single KO cells. However, since we were unable to generate an antibody capable of detecting endogenous ATP9A or ATP9B, we could not confirm these changes. Although the mechanism by which the expression of either ATP9A or ATP9B rescues the delay in VSVG transport in DKO cells remains to be elucidated, overexpression of either ATP9A or ATP9B could potentially lead to restoration.

Most mammalian P4-ATPases, except for ATP9A and ATP9B, form a heteromeric complex with CDC50A, a chaperone-like protein (Takatsu et al, 2011). Interaction of these P4-ATPases with CDC50A is necessary for their exit from the ER and localization to cellular compartments and the plasma membrane (Bryde et al, 2010; van der Velden et al, 2010; Takatsu et al, 2011). However, ATP9A or ATP9B does not require its interaction with CDC50A for their localization to the TGN or endosomes. Therefore, it is plausible to speculate that ATP9A and ATP9B form a homomeric and heteromeric complex instead of interacting with CDC50A. In SDS–PAGE, HA- or FLAG-tagged ATP9A and ATP9B migrated at ~110 and ~120 kD, respectively (Fig 3A). In BN-PAGE, ATP9A-HA and ATP9B-HA showed migration at ~470~480 and 240 kD, with ATP9B-HA slightly higher migration than ATP9A-HA (Fig 3B). These results suggest that ATP9A and ATP9B may form a tetramer and a dimer, although the possibility of their association with other proteins cannot be excluded. In addition, the upper band signal of ATP9B was stronger than the lower band signal, indicating a preference for tetramer formation. Moreover, ATP11A-HA migrated at ~400 kD, suggesting a heterotetramer, likely involving its heteromeric complex with CDC50A.

While the ATP9A-positive puncta increased in cells lacking ATP9B, the distribution of Golgin97, a TGN marker, remained consistent (Fig 4). In addition, the distribution of other proteins related to membrane trafficking, such as SNAREs and clathrin adapter proteins, was not significantly altered in cells depleted of ATP9A and ATP9B (Figs 5 and S6). Therefore, ATP9B likely contributes to the retention and/or retrieval of ATP9A to the TGN, possibly through interaction with ATP9A.

structures emerging from the Golgi apparatus, whereas green arrows indicate punctate structures in the cytoplasm. Bars, 10 $\mu$m (enlarged images, 5 $\mu$m). Images of DKO cells are shown in Fig S7B. **(B, C, D, E, F, G)** The number of cells containing membrane tubules, identified by GalT-EGFP signals, was quantified. Over 100 cells were counted per sample, except for the 30 min time points (31–59 cells) due to reduced FRB-GalT signals in many cells. **(B, C, D, E, F, G)** The graphs show the mean with ± SD from five independent experiments (B, C) and three independent experiments (D, E, F, G). Variance among individual panels (B, C, D, E, F, G) was analyzed using two-way ANOVA, followed by Dunnett's analysis for comparisons in panels (B, C) (indicated in magenta), and by Tukey's post hoc analysis for pairwise comparisons among panels (C, D, E, F, G) (indicated in blue). Individual time points in panels (C, D, E, F, G) were assessed separately by Tukey's post hoc analysis and nonsignificant results are not displayed.

 **Life Science Alliance**

ADP-ribosylation factors (ARFs) are central to vesicular transport from the Golgi apparatus. The ARF-like protein ARL1, which localizes to the TGN, plays a crucial role in Golgi-mediated membrane trafficking through the recruitment of its effectors, Arfaptins 1 and 2 (Man et al, 2011). Arfaptins possess the BAR domain, which senses membrane curvature and promotes membrane tubulation (Nakamura et al, 2012). ARL1 is essential for the exocytic pathway in *Drosophila* (Torres et al, 2014), and arfaptins are involved in the exocytic pathway, particularly in exocytic granule biogenesis (Gehart et al, 2012; Cruz-Garcia et al, 2013). Moreover, yeast Arl1p interacts with Drs2p, a flippase localizing to the Golgi apparatus (Graham, 2013; Tsai et al, 2013; Hsu et al, 2014), and exhibits genetic and biochemical links to Neo1p, an ATP9 ortholog in yeast (Wicky et al, 2004). Therefore, ATP9A/ATP9B, ARL1, and arfaptins may be functionally linked in the exocytic pathway from the TGN. However, the TGN localization of ARL1 and arfaptins is not disrupted by the depletion of either ATP9A, ATP9B, or both (Fig S6) indicating that ATP9A or ATP9B is not required for their recruitment to the Golgi. Given that BAR domain-inducible tubule formation was impaired in DKO cells (Fig 6), it is plausible that the flippase activities of ATP9A and ATP9B can facilitate membrane deformation at the Golgi apparatus, thereby contributing to the exocytic pathway.

# Materials and Methods

### Plasmids

Expression vectors encoding ATP9A and ATP9B, tagged at the C-terminus with HA- or FLAG, and their E-to-Q mutants were constructed as previously described (Takatsu et al, 2011; Tone et al, 2020). Each cDNA was transferred to pMXs-neo-DEST-HA, pMXs-blast-DEST-HA, and pMXs-puro-DEST-FLAG expression vectors using the Gateway system (Invitrogen). The pMXs-neo-vector and pEF-gag-pol plasmid were kindly provided by Toshio Kitamura (University of Tokyo). The pCMV-VSVG-Rsv-Rev plasmid (RDB04393) was a generous gift from Hiroyuki Miyoshi (RIKEN BRC). The pCAG-based vector for ATP9A expression with a C-terminal HA tag was previously described (Takatsu et al, 2011). Expression vectors for N-terminally EGFP-tagged VSVG-*ts*O45 (pCIpreEGFP-VSVG) (Nishimoto-Morita et al, 2009) and C-terminally EGFP-tagged pcDNA3-VSVG-EGFP, provided by Jennifer Lippincott-Schwartz (National Institutes of Health) (Presley et al, 1997), were transfected into HeLa cells using FuGene6 (Promega). Constructs for FKBP-conjugated fluorescent BAR (human amphiphysin 1, 26–248 a.a., N-terminal amphipathic helix is deleted) and FRB-conjugated vectors were obtained as previously detailed (Komatsu et al, 2010; Suarez et al, 2014; Takada et al, 2018). Mouse galactosyltransferase (GalT) in the pEGFP-N1 plasmid was a kind gift from Masayuki Murata (University of Tokyo). DNA fragments of FRB were amplified by PCR and inserted into the GalT-EGFP plasmid using the SLiCE cloning method (Zhang et al, 2012).

### Antibodies and reagents

The antibodies used in this study were sourced as follows: monoclonal mouse anti-EEA1 (clone 14), anti-Lamp-1 (H4A3), and anti-Golgin97 (CDF4) from BD Biosciences; monoclonal mouse anti-γ-adaptin (100.3) from Sigma-Aldrich; monoclonal mouse anti-Arfaptin2 (2B5) from Abnova; monoclonal mouse anti-DYKDDDDK from Wako; monoclonal rat anti-HA (3F10) from Roche Applied Science; polyclonal rabbit anti-Vti1a and anti-syntaxin16, from SySy; polyclonal rabbit anti-GFP from molecular probes; polyclonal goat anti-Arfaptin1 (I-19) from SantaCruz; Alexa Fluor–conjugated secondary antibodies from molecular probes; Cy3-and horseradish peroxidase–conjugated secondary antibodies from Jackson ImmunoResearch Laboratories. The polyclonal rabbit anti-Arl1 was previously described (Shin et al, 2005). The polyclonal rabbit anti-TGN46 (Kain et al, 1998) was a generous gift from Minoru Fukuda (Sanford-Burnham Medical Research Institute). Rapamycin was obtained from Sigma-Aldrich.

### Cell culture, transfection, establishment of stable cells, and immunofluorescence

HeLa cells were cultured in a minimal essential medium supplemented with 10% heat-inactivated fetal bovine serum. Parental HeLa cells and KO cells stably expressing P4-ATPases or their EQ mutants, tagged at the C-terminus with HA or FLAG, were established via retroviral infection as previously described (Takatsu et al, 2014). The infected cells were selected in a medium containing G418 (1 mg/ml), puromycin (1 μg/ml), or blasticidin (15 μg/ml). The transport of EGFP-tagged VSVG to the cell surface or to the Golgi was examined as previously detailed (Shin et al, 2005; Nishimoto-Morita et al, 2009). In brief, cells transfected with pCIpreEGFP-VSVG or pcDNA3-VSVG-EGFP were incubated at 40°C overnight, then shifted to 32°C for up to 120 min or 19.5°C for up to 60 min. Cycloheximide (150 μg/ml) was added during the incubation at 32°C and 19.5°C to inhibit protein synthesis. The former cells were stained with an anti-GFP antibody without permeabilization to detect EGFP-VSVG that reached the cell surface. HeLa cells were transfected with plasmids encoding the mCherry-FKBP-BAR domain and FRB-GalT-EGFP using FuGene6. Cells grown on coverslips were treated with 400 nM rapamycin for indicated times. Posttreatment, cells were fixed with 3% PFA in PBS at 37°C for 15 min, washed, and mounted using Mowiol. Cell observations were conducted using an AxioObserver Z1 microscope (Carl Zeiss).

### Gene editing by CRISPR/Cas9

We inactivated the *ATP9A* gene in HeLa cells using the CRISPR/Cas9 system (Tanaka et al, 2016). Target sequences for the *ATP9B* gene were designed using the CRISPR Design Tool from the Zhang lab (https://zlab.squarespace.com/guide-design-resources). Complementary oligonucleotides (5′-caccgcgggagccgaccggcacagc-3′ and 5′-aaacgctgtgccggtcggctcccgc-3′; target sequences are underlined) were synthesized and introduced into the *Bbs*I-digested PX459 vector (#48139; Addgene). To edit the *ATP9B* gene, we employed a method previously described (Tanaka et al, 2016; Nozaki et al, 2017), which involved co-transfecting the plasmid containing *ATP9B* target sequences and the Cas9 gene, and a donor plasmid (pDonor-tBFP-NLS-Neo; deposited in # 80766; Addgene), referred to as Donor (Figs S2 and S4). Both plasmids were introduced into HeLa cells via transfection using the

X-tremeGENE9 DNA Transfection Reagent (Roche). Transfected cells were selected in a medium containing G418 (1 mg/ml), and clones were isolated based on the expression of the reporter gene Tag-BFP using the SH800S cell sorter. To confirm *ATP9B* editing, genomic DNA was extracted from individual clones and subjected to PCR to amplify the region of interest. The KO was confirmed by direct sequencing of the amplified PCR product, revealing mutations in two alleles (Fig S2). The joint sequence between the donor and *ATP9B* gene was confirmed in clone 1, but not in clone 4, despite the donor being inserted into the *ATP9B* gene area of chromosome 18 (Fig S2). Clone 1 was used for further rescue experiments in this study. For *ATP9A/ATP9B*-DKO, *ATP9A*-KO (clone 23) cells were transfected with the two plasmids as described above, cells were selected in a medium containing puromycin (1 μg/ml), and clones were isolated using the SH800S cell sorter. To confirm *ATP9B* editing, genomic DNA was extracted from individual clones and subjected to PCR to amplify the region of interest. The KO was confirmed by direct sequencing of the amplified PCR product, revealing mutations in two alleles (Fig S4).

### Immunoprecipitation

Cells were lysed in lysis buffer (20 mM Hepes, pH 7.4, 150 mM NaCl, 1 mM EDTA, and 1% Nonidet P-40) containing a protease inhibitor cocktail (Nacalai Tesque) at 4°C for 30 min. The lysates were centrifuged at maximum speed for 20 min at 4°C in a microcentrifuge to remove cellular debris and insoluble materials. The supernatant was incubated at 4°C for 15 min with an anti-HA antibody and further incubated at 4°C for 16 h with protein G-coupled Dynabeads (Invitrogen). After washing, beads were incubated at 37°C for 2 h in SDS sample buffer, and the supernatant was subjected to SDS–PAGE. Immunoblot analysis was performed using rat anti-HA and mouse anti-DYKDDDDK antibodies (for FLAG tag). Immunoblots were developed using ImmunoStar (Wako) and recorded on an ImageQuant 800 instrument (Amersham).

### Immunoblot analysis

Cells were lysed in a lysis buffer (20 mM Hepes, pH 7.4, 150 mM NaCl, 1% Nonidet P-40, 1 mM EDTA) with a protease inhibitor mixture (Nacalai Tesque) at 4°C for 60 min. The lysates underwent centrifugation at maximum speed for 20 min at 4°C in a microcentrifuge to remove cellular debris and insoluble materials. Cell lysates (10 μg/sample) were incubated in SDS sample buffer containing β-mercaptoethanol at 37°C for 2 h or at RT overnight. Subsequently, the samples were subjected to SDS–PAGE and immunoblot analysis using rat anti-HA, mouse anti-actin, or mouse anti-TfnR antibodies. Immunoblots were developed using ImmunoStar reagents (Wako) and recorded on an ImageQuant 800 instrument (Amersham).

### BN-PAGE

A NativePAGE Bis-Tris Gel system (Thermo Fisher Scientific) was used for BN-PAGE (Wittig et al, 2006; Suzuki et al, 2016). Cells were lysed with a lysis buffer (20 mM Tri–HCl [pH 7.5], 50 mM KCl, 1 mM MgCl$_2$, 1% glycerol, 1% n-dodecyl-β-D-maltopyranoside [DDM], protease inhibitor cocktail [Nacalai Tesque]) on ice for 1 h. Cell

lysates (~15 μg) were loaded onto NativePAGE 4–16% Bis–Tris Gels. After electrophoresis at 150 V at 4°C for 30 min, the concentration of Coomassie brilliant blue G-250 in the cathode running buffer was reduced from 0.02% to 0.002%, and the sample was further electrophoresed at 150 V for 90 min. Protein standards (Thermo Fisher Scientific) were used as molecular weight markers. The gels were soaked in SDS–PAGE running buffer (25 mM Tris, 192 mM glycine, and 0.1% SDS) at room temperature for 20 min and then transferred to PVDF membranes (Immobilon-P; Millipore). Membranes were probed with anti-HA or anti-DYKDDDK antibodies, followed by incubation with HRP-conjugated anti-rat or anti-mouse antibodies. Immunoblots were developed using ImmunoStar reagents (Wako) and recorded on an ImageQuant 800 instrument (Amersham).

## Data Availability

The raw data for all experiments are available from the corresponding author upon reasonable request.

## Supplementary Information

## Acknowledgements

We thank Jun Suzuki and Niu Han (Kyoto University) for their technical support with BN-PAGE, and Toshio Kitamura and Masayuki Murata (The University of Tokyo), Hiroyuki Miyoshi (RIKEN BioResource Center), Minoru Fukuda (Sanford-Burnham Medical Research Institute), and Peter McPherson (McGill University) for providing materials. This work was supported by JSPS KAKENHI Grant Numbers JP23H02434 and JP23K27127 (to H-W Shin), the Takeda Science Foundation (to H-W Shin and Y Katoh), the Mizutani Glycoscience Foundation (to H-W Shin), and the ONO Medical Research Foundation (to H-W Shin).

### Author Contributions

T Yagi: data curation, formal analysis, validation, investigation, visualization, methodology, and writing—review and editing.
R Nakabuchi: data curation, formal analysis, investigation, methodology, and writing—review and editing.
Y Muranaka: resources, formal analysis, and methodology.
G Tanaka: resources and methodology.
Y Katoh: resources.
K Nakayama: validation and writing—review and editing.
H Takatsu: resources, data curation, validation, investigation, visualization, methodology, and writing—review and editing.
H-W Shin: conceptualization, resources, formal analysis, supervision, validation, project administration, and writing—original draft, review, and editing.

### Conflict of Interest Statement

The authors declare that they have no conflict of interest.

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
