## [Reviewer comments · Life Science Alliance]

Life Science Alliance

Lipid flippases ATP9A and ATP9B form a complex and contribute to the exocytic pathway from the Golgi

Tsukasa Yagi, Riki Nakabuchi, Yumeka Muranaka, Gaku Tanaka, Yohei Katoh, Kazuhisa Nakayama, Hiroyuki Takatsu, and Hye-Won Shin

DOI: <https://doi.org/10.26508/lsa.202403163>

Corresponding author(s): *Hye-Won Shin, Kyoto University*

Review Timeline:

Submission Date:	2024-12-06
Editorial Decision:	2025-01-16
Revision Received:	2025-03-14
Editorial Decision:	2025-03-26
Revision Received:	2025-03-26
Editorial Decision:	2025-03-28
Revision Received:	2025-03-31
Accepted:	2025-03-31

Scientific Editor: *Eric Sawey, PhD*

Transaction Report:

January 16, 2025

Re: Life Science Alliance manuscript #LSA-2024-03163-T

Dr. Hye-Won Shin
Kyoto University
Graduate School of Pharmaceutical Sciences
46-29 Yoshida-shimo-adachi-cho
Sakyo-ku
Kyoto, Kyoto 606-8501
Japan

Dear Dr. Shin,

Thank you for submitting your manuscript entitled "Lipid flippases ATP9A and ATP9B form a complex and contribute to the exocytic pathway from the Golgi" to Life Science Alliance. The manuscript was assessed by expert reviewers, whose comments are appended to this letter. We invite you to submit a revised manuscript addressing the Reviewer comments.

Thank you for this interesting contribution to Life Science Alliance. We are looking forward to receiving your revised manuscript.

Sincerely,

B. MANUSCRIPT ORGANIZATION AND FORMATTING:

Reviewer #1 (Comments to the Authors (Required)):

Following up their previous work, Yagi and colleagues now show that the flippases ATP9A and ATP9B likely form homomeric and/or heteromeric complexes (dimers and tetramers) and that heteromeric complexes may also contribute to intracellular distribution of ATP9A in the TGN. Importantly, their ATPase/flippase activities are required for the efficient secretion of the VSVG protein. Although the double KO of both flippases delays VSVG secretion, it apparently does not perturb the overall organization of the Golgi/endosomal compartments. Experiments using a recombinant curvature sensor/inducer construct (containing amphiphysin) further suggest that the flippase activity contributes to the formation of tubular/vesicular structures. Overall, the manuscript is well written and the work further establishes a role of the P4 type ATP9A/B flippases in protein secretion. A surprising finding is that HA-tagged ATP9A or ATP9B rescue the delay in VSVG transport in the ATP9A and ATP9B double KO. In contrast, in the single ATP9A and ATP9B KOs, which should still contain the other endogenous ATP9 isoform, the delay in secretion is apparent. This unexpected finding needs explanation and in particular it should be tested to which degree the expression levels of the endogenous (remaining) isoform and of the exogenous HA-tagged isoforms differ. For example, does the ATP9A KO also affect/reduce the endogenous ATP9B level (and vice versa). In other words, are the endogenous amounts of ATP9 rate-limiting for efficient VSVG secretion?

The authors should also address the following issues:

- In addition to the delayed cell surface appearance of EGFP-VSVG, do the authors also observe an accumulation of VSVG in a specific post-Golgi compartment at extended chase periods at 32°C?
- Figure 1C: Compared to WT, the expression level of the ATP9B(E272Q) construct in the ATP9B KO is much lower. Thus, it is difficult to conclude that the ATPase-deficient mutant rescues the VSVG transport to lesser degree than the WT ATP9B flippase. Please show data with cells containing similar expression levels of WT and mutant (E272Q) ATP9B.
- Figure 3A, lane 5: Compared to the other HA-tagged ATP9A immunoprecipitations (lanes 1-4,) the IP, shown in lane 5, seems to be less efficient and in addition the expression levels of ATP11C in the total lysate are low. Thus, it is not clear, whether ATP9A does not bind ATP11C. This experiment should be repeated, showing a more efficient HA-ATP9A and if possible using cell show higher ATP11C expression levels. Furthermore, Figure 3B could also include controls, coexpressing HA-tagged ATP11 together with either FLAG-tagged ATP9A or ATP9B, demonstrating that the position of the HA-ATP11-band is not shifted in the presence of ATP9A/B.
- Figure 4A: Please also mention the TfnR and EEA1 staining in the legend.
- Please explain abbreviations (e.g. CBB).

Reviewer #2 (Comments to the Authors (Required)):

The manuscript by Yagi and colleagues are exploring the role of human flippases ATP9A and ATP9B in the exocytic pathway. Firstly, the authors created the ATP9A and ATP9B knockout in the HeLa Cells and tracked the classical VSVG transport from ER-Golgi to the plasma membrane. Authors showed that ATP9A and 9B Knockout cells showed delayed VSVG transport and their flippase activity is essential for the VSVG transport. Interestingly, authors claim that ATP9A and ATP9B functions are redundant and form a homomeric or heteromeric complex. Furthermore, the authors used the chemically inducible dimerization (CID) method to show that ATP9A and 9B are important for membrane deformation. Overall, the manuscript presents interesting observations but lacks evidence for concluding that ATP9A and 9B play a role in the exocytic pathway. I have suggestions for improving the manuscript.

1. Fig S1A: Localization of ATP9A, ATP9B: A recent study from Meng et al. 2023, carefully checked the localization of ATP9A with different organelle markers and found that ATP9A only localizes to the endosomal compartment and does not localize to the Golgi. I think to clarify these discrepancies authors need to carefully check the colocalization of the ATP9A and 9B with different organelles (Golgi, Endosomes) and quantify the colocalization coefficient (Pearson or Manders).
2. Fig S2: Authors need to verify the ATP9A and ATP9B knockout using immunoblotting or quantitative PCR.

3. Fig1A: Authors claimed that 9A/9B knockout delays the VSVG transport from the Golgi to the plasma membrane but to claim the delay authors need to perform a kinetics analysis of VSVG transport.
4. How did the authors quantify the surface fluorescence intensity? Is it possible to use any surface marker or a plasma membrane marker to quantify the VSVG localization on the surface?
5. VSVG transport: Authors need to quantify the total VSVG fluorescence and then compare the VSVG fluorescence at the Golgi and surface. This data will help to understand the distribution of VSVG fluorescence. I don't see any changes in the Golgi VSVG in the control cell or VSVG accumulation in the 9A/9B cells.
6. I wonder if the authors can use any other cargo and test whether it shows similar transport defects or if it is VSVG-specific. An example of a similar system could be a RUSH system.
7. Fig 1C: Why is there a low expression of ATP9B(EQ) in 9B KO cells and surprisingly it can still suppress the VSVG transport defects?
8. Fig 2A, B: I assume that control cells (wild type HeLa Cells) express endogenous ATP9A and 9B but in the Fig 2B quantification the VSVG surface intensity is low compared to Fig1B and interestingly 9A and 9B complementation increases the VSVG surface intensity compared to the control cells? Is it a significant difference? Authors need to confirm the expression of ATP9A, 9B in the control cells vs overexpression cells.
9. Authors suggest the redundant function of ATP9A and 9B but it still confuses me that in Fig 1A, a single knockout of either 9A or 9B causes a defect in VSVG transport but in DKO cells expression of only 9A or 9B can compensate for VSVG transport defects and suppress it.
10. Fig 3A: It appears that ATP9A (EQ), ATP9B WT shows reduced interaction with ATP9A but surprisingly ATP9B EQ interact strongly with ATP9A. A quantification of the % CO-IP of the different interacting proteins would clarify this issue.
11. It's interesting that 9A and 9B form homomeric and heteromeric complexes. The authors need to discuss a possible explanation of homomeric and heteromeric complexes for flippases. Is it essential for flippase activity? Yeast protein Neo1 exists in a monomeric form and shows flippase activity.
12. Fig 4A: It would be more appropriate to show changes in ATP9A distribution by quantifying the colocalization of ATP9A with Golgin 97 and endosomal markers.
13. interestingly, 9A and 9B flippase activities are required for membrane deformation but it seems an incomplete story. Authors need to test some cargo that may be affected by this process. Is it related to endosomal recycling or specific for exocytic cargos?
14. Line 341-342, " While ATP9A and ATP9B can form a homomeric complex in ATP9B-KO and ATP9A-KO cells, respectively, VSVG transport to the plasma membrane was delayed in these KO cells. This suggests that the heteromeric complex may be more efficient in facilitating VSVG transport" The quantification in Fig 1A, B does not represent this statement.

RE: LSA-2024-03163-T

Newly added Figure

Supplementary Figure S6 is added.

Modification of existing Figures

Figure 1C is exchanged with new data.

Supplementary Figure S3A is added.

Supplementary S6 is renamed to S7.

Supplementary S7 is renamed to S8.

Reviewer #1 (Comments to the Authors (Required)):

Following up their previous work, Yagi and colleagues now show that the flippases ATP9A and ATP9B likely form homomeric and/or heteromeric complexes (dimers and tetramers) and that heteromeric complexes may also contribute to intracellular distribution of ATP9A in the TGN. Importantly, their ATPase/flippase activities are required for the efficient secretion of the VSVG protein. Although the double KO of both flippases delays VSVG secretion, it apparently does not perturb the overall organization of the Golgi/endosomal compartments. Experiments using a recombinant curvature sensor/inducer construct (containing amphiphysin) further suggest that the flippase activity contributes to the formation of tubular/vesicular structures.

Overall, the manuscript is well written and the work further establishes a role of the P4 type ATP9A/B flippases in protein secretion. A surprising finding is that HA-tagged ATP9A or ATP9B rescue the delay in VSVG transport in the ATP9A and ATP9B double KO. In contrast, in the single ATP9A and ATP9B KOs, which should still contain the other endogenous ATP9 isoform, the delay in secretion is apparent. This unexpected finding needs explanation and in particular it should be tested to which degree the expression levels of the endogenous (remaining) isoform and of the exogenous HA-tagged isoforms differ. For example, does the ATP9A KO also affect/reduce the endogenous ATP9B level (and vice versa). In other words, are the endogenous amounts of ATP9 rate-limiting for efficient VSVG secretion?

We greatly appreciate the reviewer's constructive and positive comments. As the reviewer noted, it was unexpected that the expression of either ATP9A or ATP9B could rescue the delayed VSVG transport in DKO cells, while the delay was still observed in single KO cells. Unfortunately, despite multiple attempts, we were unable to generate an antibody capable of detecting endogenous ATP9A or ATP9B. As a result, we could not

assess changes in endogenous ATP9A or ATP9B levels in single KO cells. We added relevant sentences on page 13 in the discussion.

The authors should also address the following issues:

- In addition to the delayed cell surface appearance of EGFP-VSVG, do the authors also observe an accumulation of VSVG in a specific post-Golgi compartment at extended chase periods at 32°C?

VSVG transport to the plasma membrane was not inhibited but delayed. Consequently, longer incubation leads to a decrease in EGFP-VSVG signals in the Golgi in both control and KO cells. Due to the variability in EGFP-VSVG expression levels from transient expression, it is challenging to accurately assess changes in the remaining EGFP-VSVG signals in the Golgi after prolonged incubation. At early time points, the Golgi-signal of EGFP-VSVG is prominent, while only a minimal amount of EGFP-VSVG is transported to the plasma membrane. In this study, we detected this low level of EGFP-VSVG by immunofluorescence without permeabilization. As we described in the original manuscript, to minimize the variability in EGFP-VSVG expression, we normalized the surface EGFP-VSVG signal (red fluorescence) to the EGFP-VSVG signal (green fluorescence), which predominantly localized in the Golgi.

- Figure 1C: Compared to WT, the expression level of the ATP9B(E272Q) construct in the ATP9B KO is much lower. Thus, it is difficult to conclude that the ATPase-deficient mutant rescues the VSVG transport to lesser degree than the WT ATP9B flippase. Please show data with cells containing similar expression levels of WT and mutant (E272Q) ATP9B.

We have replaced Figure 1C with new data.

- Figure 3A, lane 5: Compared to the other HA-tagged ATP9A immunoprecipitations (lanes 1-4,) the IP, shown in lane 5, seems to be less efficient and in addition the expression levels of ATP11C in the total lysate are low. Thus, it is not clear, whether ATP9A does not bind ATP11C. This experiment should be repeated, showing a more efficient HA-ATP9A and if possible using cell show higher ATP11C expression levels. Furthermore, Figure 3B could also include controls, coexpressing HA-tagged ATP11 together with either FLAG-tagged ATP9A or ATP9B, demonstrating that the position of the HA-ATP11-band is not shifted in the presence of ATP9A/B.

Despite multiple attempts to generate cells stably expressing ATP11C, its expression level remained comparable to that of the original cells. Based on our experience, ATP9A

and ATP9B exhibit higher expression levels than other P4-ATPases. We believe that the expression level of ATP11C is sufficient for detection when co-precipitated with ATP9A or ATP9B. Supporting this, the migration of ATP11A in the BN-PAGE was unchanged between cells expressing ATP11A alone and those coexpressing ATP11A with ATP9 (newly added Supplementary Figure S6).

- Figure 4A: Please also mention the TfnR and EEA1 staining in the legend.

We apologize for not including information on TfnR and EEA1. We have now added this information to the Figure 4 legend.

- Please explain abbreviations (e.g. CBB).

We added the full name of CBB (Coomassie brilliant blue) to page 19.

Reviewer #2 (Comments to the Authors (Required)):

The manuscript by Yagi and colleagues are exploring the role of human flippases ATP9A and ATP9B in the exocytic pathway. Firstly, the authors created the ATP9A and ATP9B knockout in the Hela Cells and tracked the classical VSVG transport from ER-Golgi to the plasma membrane. Authors showed that ATP9A and 9B Knockout cells showed delayed VSVG transport and their flippase activity is essential for the VSVG transport. Interestingly, authors claim that ATP9A and ATP9B functions are redundant and form a homomeric or heteromeric complex. Furthermore, the authors used the chemically inducible dimerization (CID) method to show that ATP9A and 9B are important for membrane deformation. Overall, the manuscript presents interesting observations but lacks evidence for concluding that ATP9A and 9B play a role in the exocytic pathway. I have suggestions for improving the manuscript.

We appreciate the reviewer's constructive comments.

1. Fig S1A: Localization of ATP9A, ATP9B: A recent study from Meng et al. 2023, carefully checked the localization of ATP9A with different organelle markers and found that ATP9A only localizes to the endosomal compartment and does not localize to the Golgi. I think to clarify these discrepancies authors need to carefully check the colocalization of the ATP9A and 9B with different organelles (Golgi, Endosomes) and quantify the colocalization coefficient (Pearson or Manders).

Meng et al. indeed showed that ATP9A is highly expressed in endocytic organelles in

neuroblastoma (Neuro2A) cells. We and other demonstrated that ATP9A is localized to the endosomes and the *trans*-Golgi network in HeLa cells (Takatsu et al., 2011; Tanaka et al., 2016; McGough et al., 2018). While we do not have a clear explanation for the differences in subcellular localization between in HeLa and Neuro2A cells, the localization of some P4-ATPases appears to vary across different cell types (Shin et al. 2020). For example, ATP8A1, a P4-ATPases member, localizes to recycling endosomes in COS-1 cells (Lee et al., 2015), to late endosomes in HeLa and alveolar type 2 cells (Takatsu et al., 2011; Kook et al., 2021; Lan et al., 2025), to the Golgi and the plasma membrane in U2OS cells (van der Velden et al., 2010), and recycling endosomes and the plasma membrane in CHO-1 cells (Kato et al., 2013).

2. Fig S2: Authors need to verify the ATP9A and ATP9B knockout using immunoblotting or quantitative PCR.

Unfortunately, despite multiple attempts, we were unable to generate an antibody capable of detecting endogenous ATP9A or ATP9B, preventing us from measuring their endogenous levels. Since the CRISPR-Cas9 system edits the genome, quantitative RT-PCR cannot be used to compare expression levels.

3. Fig1A: Authors claimed that 9A/9B knockout delays the VSVG transport from the Golgi to the plasma membrane but to claim the delay authors need to perform a kinetics analysis of VSVG transport.

In this experiment, the delay at early time points is evident, although a kinetic analysis could not be performed. To address this, we would need to establish another system such as RUSH, as the reviewer suggested. However, the delay in KO cells is clearly demonstrated through control and rescue experiments. Therefore, we did not conduct a kinetic analysis and hope the reviewer finds this acceptable.

4. How did the authors quantify the surface fluorescence intensity? Is it possible to use any surface marker or a plasma membrane marker to quantify the VSVG localization on the surface?

We apologize for the lack of sufficient information about our experimental system. We have now added a new illustration in Supplementary Figure S3A.

5. VSVG transport: Authors need to quantify the total VSVG fluorescence and then compare the VSVG fluorescence at the Golgi and surface. This data will help to understand the distribution of VSVG fluorescence. I don't see any changes in the Golgi

VSVG in the control cell or VSVG accumulation in the 9A/9B cells.

VSVG transport to the plasma membrane was not inhibited but delayed, with the delay at early time points being evident. Consequently, longer incubation leads to a decrease in EGFP-VSVG signals in the Golgi in both control and KO cells. Due to the variability in EGFP-VSVG expression levels from transient expression, it is challenging to accurately assess changes in the remaining EGFP-VSVG signals in the Golgi after prolonged incubation. At early time points, the Golgi-signal of EGFP-VSVG is prominent, while only a minimal amount of EGFP-VSVG is transported to the plasma membrane.

In this study, to enhance the detection of surface EGFP-VSVG signals, we incubated the cells with anti-GFP and followed by secondary antibodies without permeabilization. As we described in the original manuscript, to minimize the variability in EGFP-VSVG expression, we normalized the surface EGFP-VSVG signal (red fluorescence) to the EGFP-VSVG signal (green fluorescence), which predominantly localized in the Golgi.

6. I wonder if the authors can use any other cargo and test whether it shows similar transport defects or if it is VSVG-specific. An example of a similar system could be a RUSH system.

We appreciate the reviewer's comments and agree that analyzing other cargo is important. We are comparing several secretory pathways, including the transport of GPI-anchored cargoes. We plan to continue these experiments as part of our future work.

7. Fig 1C: Why is there a low expression of ATP9B(EQ) in 9B KO cells and surprisingly it can still suppress the VSVG transport defects?

We have replaced Figure 1C with new data.

8. Fig 2A, B: I assume that control cells (wild type HeLa Cells) express endogenous ATP9A and 9B but in the Fig 2B quantification the VSVG surface intensity is low compared to Fig1B and interestingly 9A and 9B complementation increases the VSVG surface intensity compared to the control cells? Is it a significant difference? Authors need to confirm the expression of ATP9A, 9B in the control cells vs overexpression cells. Since the experiments for Figure 1 and Figure 2 were conducted separately, the fluorescence intensities between the two figures cannot be directly compared.

9. Authors suggest the redundant function of ATP9A and 9B but it still confuses me that in Fig 1A, a single knockout of either 9A or 9B causes a defect in VSVG transport but in

DKO cells expression of only 9A or 9B can compensate for VSVG transport defects and suppress it.

As the reviewer noted, it was unexpected that the expression of either ATP9A or ATP9B could rescue the delayed VSVG transport in DKO cells, while the delay was still observed in single KO cells. While we do not have the correct answer to this, we added relevant sentences to the discussion (pages 13-14).

10. Fig 3A: It appears that ATP9A (EQ), ATP9B WT shows reduced interaction with ATP9A but surprisingly ATP9B EQ interact strongly with ATP9A. A quantification of the % CO-IP of the different interacting proteins would clarify this issue.

There is some tendency for the EQ mutant to interact more strongly with WT, potentially forming a more stable EQ-WT complex compared to the WT-WT complex. However, since the Co-IP data is not quantitative, we did not emphasize this observation in the manuscript.

11. It's interesting that 9A and 9B form homomeric and heteromeric complexes. The authors need to discuss a possible explanation of homomeric and heteromeric complexes for flippases. Is it essential for flippase activity? Yeast protein Neo1 exists in a monomeric form and shows flippase activity.

We appreciate the reviewer's comments and agree on the importance of analyzing the distinct functions of homomeric and heteromeric complexes. We are currently comparing secretory pathways, including the transport of GPI-anchored proteins, as well as the endocytic and recycling pathways. We plan to continue these experiments as part of our future work.

12. Fig 4A: It would be more appropriate to show changes in ATP9A distribution by quantifying the colocalization of ATP9A with Golgin 97 and endosomal markers.

We apologize for any confusion caused by the description in our original manuscript. As stated in our response to comment 1, ATP9A localizes the TGN and endosomes in HeLa cells. it colocalizes with the TGN and early/recycling endosomal markers but not with late endosomal markers (Takatsu et al., 2011; Tanaka et al., 2016). While ATP9A-positive puncta increased in ATP9B-KO cells, they showed only partial colocalization with TfnR- and EEA1-positive compartments. Therefore, instead of performing a colocalization analysis, we quantified the ATP9A-positive puncta in the cytoplasm in Figure 4C and D. We think the increased ATP9A-puncta include not only endocytic but also exocytic vesicular structures. Accordingly, we revised the sentences

on pages 9 and 14 to enhance clarity and avoid confusion.

13. interestingly, 9A and 9B flippase activities are required for membrane deformation but it seems an incomplete story. Authors need to test some cargo that may be affected by this process. Is it related to endosomal recycling or specific for exocytic cargos?

We appreciate the reviewer's comments. We described this point in our response to comment 11 above.

14. Line 341-342, " While ATP9A and ATP9B can form a homomeric complex in ATP9B-KO and ATP9A-KO cells, respectively, VSVG transport to the plasma membrane was delayed in these KO cells. This suggests that the heteromeric complex may be more efficient in facilitating VSVG transport" The quantification in Fig 1A, B does not represent this statement.

As we stated in our response to comment 9 above, this is a possible explanation for the unexpected results.

March 26, 2025

Re: Life Science Alliance manuscript #LSA-2024-03163-TR

Dr. Hye-Won Shin
Kyoto University
Graduate School of Pharmaceutical Sciences
46-29 Yoshida-shimo-adachi-cho
Sakyo-ku
Kyoto, Kyoto 606-8501
Japan

Dear Dr. Shin,

Thank you for submitting your revised manuscript entitled "Lipid flippases ATP9A and ATP9B form a complex and contribute to the exocytic pathway from the Golgi" to Life Science Alliance. The manuscript has been seen by the original reviewers whose comments are appended below. While the reviewers continue to be overall positive about the work in terms of its suitability for Life Science Alliance, some important issues remain.

Our general policy is that papers are considered through only one revision cycle; however, given that the suggested changes are relatively minor, we are open to one additional short round of revision. Please note that I will expect to make a final decision without additional reviewer input upon re-submission.

Please submit the final revision within one month, along with a letter that includes a point by point response to the remaining reviewer comments.

To upload the revised version of your manuscript, please log in to your account: <https://lsa.msubmit.net/cgi-bin/main.plex>
You will be guided to complete the submission of your revised manuscript and to fill in all necessary information.

B. MANUSCRIPT ORGANIZATION AND FORMATTING:

Sincerely,

Reviewer #1 (Comments to the Authors (Required)):

In the revised manuscript, Yagi and colleagues have addressed some of the previous issues. However, as also noted by the other reviewer, it is surprising that single ATP9A or ATP9B KO cells show delayed VSVG transport, but the rescue of the DKO by either ATP9A or ATP9B does not result in obvious transport delays. As already mentioned previously, a possible explanation would be that the available amounts of either ATP9A or ATP9B become "rate-limiting", thereby affecting VSVG transport efficiency. (Alternatively, this unexpected finding could be caused by indirect effects.) Two approaches could directly address this issue. Determination of the ATP9 expression levels using specific antibodies or transformation of e.g. ATP9A KO cells by exogenous ATP9B to increase the protein levels (or vice versa expression of ATP9A in ATP9B KO cells). The authors mention that they were unable to generate antibodies capable of detecting endogenous ATP9A or ATP9B. However, it appears that antibodies directed against ATP9A and ATP9B are available from commercial sources. Based on the available information, e.g. the polyclonal anti-ATP9A antibodies detect ATP9A from several species and seem to be suitable for Western blot analysis and immunofluorescence labeling. These antibodies could also be used to confirm successful KO of ATP9A/B.

Reviewer #2 (Comments to the Authors (Required)):

The revised version of this manuscript addressed all my concerns. I have no further suggestions for this manuscript

Referee Cross-Comments

The revised manuscript addresses some of my concerns. I agree with the other reviewer that the authors should confirm ATP9A/B expression for complementation assays in DKO cells and validate the knockout cells. Commercially available ATP9A antibodies have been used in previous studies by Meng et al. (2023) and Wang et al. (2023).

Reviewer #1 (Comments to the Authors (Required)):

In the revised manuscript, Yagi and colleagues have addressed some of the previous issues. However, as also noted by the other reviewer, it is surprising that single ATP9A or ATP9B KO cells show delayed VSVG transport, but the rescue of the DKO by either ATP9A or ATP9B does not result in obvious transport delays. As already mentioned previously, a possible explanation would be that the available amounts of either ATP9A or ATP9B become "rate-limiting", thereby affecting VSVG transport efficiency. (Alternatively, this unexpected finding could be caused by indirect effects.) Two approaches could directly address this issue. Determination of the ATP9 expression levels using specific antibodies or transformation of e.g. ATP9A KO cells by exogenous ATP9B to increase the protein levels (or vice versa expression of ATP9A in ATP9B KO cells). The authors mention that they were unable to generate antibodies capable of detecting endogenous ATP9A or ATP9B. However, it appears that antibodies directed against ATP9A and ATP9B are available from commercial sources. Based on the available information, e.g. the polyclonal anti-ATP9A antibodies detect ATP9A from several species and seem to be suitable for Western blot analysis and immunofluorescence labeling. These antibodies could also be used to confirm successful KO of ATP9A/B.

We greatly appreciate the reviewer's constructive comments and suggestions. We tested several commercial antibodies for ATP9A and ATP9B including monoclonal mouse anti-ATP9A (M02, Abnova), polyclonal rabbit anti-ATP9A (Sigma), and polyclonal rabbit anti-ATP9B (Santa Cruz). However, none of them worked properly in HeLa cells. While we did not test all commercially available antibodies including the abcam antibody suggested by Reviewer 2, which reported to work in neuroblastoma cells (Meng et al., 2023, Mol Psychiatry). We have not made any revisions and hope the reviewer finds this acceptable.

March 28, 2025

RE: Life Science Alliance Manuscript #LSA-2024-03163-TRR

Dr. Hye-Won Shin
Kyoto University
Graduate School of Pharmaceutical Sciences
46-29 Yoshida-shimo-adachi-cho
Sakyo-ku
Kyoto, Kyoto 606-8501
Japan

Dear Dr. Shin,

Thank you for submitting your revised manuscript entitled "Lipid flippases ATP9A and ATP9B form a complex and contribute to the exocytic pathway from the Golgi". We would be happy to publish your paper in Life Science Alliance pending final revisions necessary to meet our formatting guidelines.

- please be sure that the authorship listing and order is correct
- please add the X and Bluesky handles of your host institute/organization as well as your own or/and one of the authors in our system
- please add an Author Contributions section to your main manuscript text
- please add a Conflict of Interest statement to your main manuscript text
- please use the [10 author names, et al.] format in your references (i.e., limit the author names to the first 10)
- please add callouts for Figures S1B and S3D to your main manuscript text
- Please upload all figure files as individual ones, including the supplementary figure files; all figure legends should only appear in the main manuscript file

FIGURE CHECK:

- please add scale bars to Figure S3B, C and D

A. FINAL FILES:

B. MANUSCRIPT ORGANIZATION AND FORMATTING:

Sincerely,

March 31, 2025

RE: Life Science Alliance Manuscript #LSA-2024-03163-TRRR

Dr. Hye-Won Shin
Kyoto University
Graduate School of Pharmaceutical Sciences
46-29 Yoshida-shimo-adachi-cho
Sakyo-ku
Kyoto, Kyoto 606-8501
Japan

Dear Dr. Shin,

Thank you for submitting your Research Article entitled "Lipid flippases ATP9A and ATP9B form a complex and contribute to the exocytic pathway from the Golgi". It is a pleasure to let you know that your manuscript is now accepted for publication in Life Science Alliance. Congratulations on this interesting work.

DISTRIBUTION OF MATERIALS:

Again, congratulations on a very nice paper. I hope you found the review process to be constructive and are pleased with how the manuscript was handled editorially. We look forward to future exciting submissions from your lab.

Sincerely,
